# The Prolonged Treatment of *Salmonella enterica* Strains with Human Serum Effects in Phenotype Related to Virulence

**DOI:** 10.3390/ijms24010883

**Published:** 2023-01-03

**Authors:** Bożena Futoma-Kołoch, Michał Małaszczuk, Kamila Korzekwa, Małgorzata Steczkiewicz, Andrzej Gamian, Gabriela Bugla-Płoskońska

**Affiliations:** 1Department of Microbiology, Faculty of Biological Sciences, University of Wrocław, 51-148 Wrocław, Poland; 2Laboratory of Medical Microbiology, Hirszfeld Institute of Immunology and Experimental Therapy, Polish Academy of Sciences, Weigla 12, 53-114 Wrocław, Poland

**Keywords:** *Salmonella enterica*, human serum, resistance, C3c, C1q, antibiotics, quaternary ammonium salt, biofilm, hydrophobicity, motility, membrane proteins

## Abstract

*Salmonella enterica* as common pathogens of humans and animals are good model organisms to conduct research on bacterial biology. Because these bacteria can multiply in both the external environments and in the living hosts, they prove their wide adaptability. It has been previously demonstrated that prolonged exposition of *Salmonella* serotype O48 cells to normal human serum led to an increase in resistance to sera in connection with the synthesis of very long O-antigen. In this work, we have studied the phenotype connected to virulence of *Salmonella enterica* strains that were subjected to consecutive passages in 50% human serum from platelet-poor plasma (SPPP). We found that eight passages in SPPP may not be enough for the bacteria to become serum-resistant (*S*. Typhimurium ATCC 14028, *S*. Senftenberg). Moreover, C1q and C3c complement components bound to *Salmonellae* (*S*. Typhimurium ATCC 14028, *S*. Hammonia) membrane proteins, which composition has been changed after passaging in sera. Interestingly, passages in SPPP generated genetic changes within gene *fljB*, which translated to cells’ motility (*S*. Typhimurium ATCC 14028, *S*. Erlangen). One strain, *S*. Hammonia exposed to a serum developed a multi-drug resistance (MDR) phenotype and two *S*. Isaszeg and *S*. Erlangen tolerance to disinfectants containing quaternary ammonium salts (QAS). Furthermore, colonial morphotypes of the serum adaptants were similar to those produced by starter cultures. These observations suggest that overcoming stressful conditions is manifested on many levels. Despite great phenotypic diversity occurring after prolonged exposition to SPPP, morphotypes of colonies remained unchanged in basic media. This work is an example in which stable morphotypes distinguished by altered virulence can be confusing during laboratory work with life-threatening strains.

## 1. Introduction

Despite the promotion of hygiene and the prevention of diseases by the maintenance of sanitary conditions, the transmission of *Salmonella* spp. continues to affect communities worldwide [1] According to the Center for Disease Control and Prevention, *Salmonella* alone affects about 1.4 million people yearly in the United States with about 16,000 hospitalizations and more than 500 deaths annually [2]. In the EU, salmonellosis was the second most reported zoonotic infection in 2009, with 108,614 cases confirmed and a fatality rate of 0.08%, which approximately corresponds to 90 human deaths [3]. Infections caused by these microorganisms are self-limiting and not usually life-threatening; however, in infants, children, adults, transplant recipients, pregnant women, and people with weakened immune systems, the development of complications can be dangerous [4]. The endocardium, arteries, central nervous system, lungs, bones, joints, muscles, soft tissues, reticuloendothelial system, kidneys, and genital regions have all been documented sites of extraintestinal infection [5]. Most cases of bacteremia occur in Africa and are caused by non-typhoidal *Salmonella* (NTS) strains in patients infected with HIV [6]. *Salmonella* Enteritidis and *Salmonella* Typhimurium are the most frequently isolated serotypes from all countries submitting data [7]. In Poland, *Salmonella* Enteritidis is noted in most of the food poisonings. These pathogens are found mainly in meat and dairy products, poultry, eggs, and their products and in fish, cottage cheese, cheese hard, soft, and melted, and fruit juices. Serovar Senftenberg has been detected in the environment and production lines. It has been identified as an etiological factor of invasive intestinal infections in humans. Its presence is a serious problem, especially in the poultry industry [8]. 

Within a great number of *Salmonella* serotypes (more than 2500), there is a unique serogroup O48 which comprises strains containing sialic acid (NeuAc) in O-antigen polysaccharide (O-Ag) of lipopolysaccharide (LPS) [9]. It has been also shown that strains differ in NeuAc/Kdo ratio which provides information about the amount of NeuAc residues per LPS molecule, from which the average length of the O-specific part of the LPS of the strain can be deduced. It has been found that *Salmonella* Hammonia had a very high NeuAc/Kdo ratio, *Salmonella* Erlangen a medium level, and the NeuAc content of *Salmonella* Isaszeg PCM 2550 was below the detection limit [10].

The incorporation of NeuAc into LPS is thought to be a strategy of defense against serum complement (C) activation [11]. LPS is classified as the key virulence factor and a major surface component of *Salmonella enterica* [12]. The presence of very long O-antigen (VL-OAg) in the NTS protects bacteria from the lytic action of serum complement [13], although the relationship between antibody, O-Ag, and other outer-membrane antigens is poorly understood. LPS itself protects microbes from C lytic action in the way of impaired access of membrane attack complex (MAC) to the outer membrane (OM) [14]; however, it has been demonstrated that some *Salmonella* O48 serovars that possessed the smooth type of LPS (S form, composed of lipid A, core and O-Ag with NeuAc) were sensitive to normal human serum (NHS) (Figure 1) [10,15]. Additionally, prolonged exposure of *Salmonella* O48 cells to NHS (multiple passages) led to an increase in resistance to NHS in connection with the synthesis of very VL-OAg (more than 100 O-specific units) [16,17]. As other sources show, the resistance of *Salmonella* rods to sera can also be conditioned by the presence of some classes of outer membrane proteins (OMPs). So far, four OMPs Rck, PagC, PgtE and TraT have been described, which enable *Salmonella* to survive in serum [17]. We previously reported that in serum-resistant (SR) and quaternary ammonium salts (QAS) tolerant *Salmonella* Senftenberg strain, flagellar protein FliC, as well as enolase were downregulated in comparison to the wild-type (serum sensitive, SS and QAS-sensitive) parental strains. In contrast, chemotaxis response regulator protein-glutamate methylesterase and outer membrane protein assembly factor BamA were upregulated (Figure 1) [18]. 

The central event of the C activation cascade is the activation of the C3 protein, which drives the biological effects of complement. C3 molecule supports the activation of the three pathways of the C cascade: classical (CP), lectin (LP), and alternative (AP). The pivotal step in C activation is the conversion of the C3 component to C3b, C3f, iC3b, C3c, C3dg, C3g, and C3d [19,20,21]. So far, deposition of C3c fragments using ELISA assay has been investigated on three *Salmonella* O48 strains (*S*. Isaszeg, *S*. *arizonae*, *S*. Ngozi) of SS phenotype and it has been found that the highest C3c binding occurred on *S*. Isaszeg (statistical significance *p* < 0.05). Considering the involvement of OMPs in the susceptibility to serum, we found that binding of C3c occurred on the OMPs of three tested isolates in the range of molecular masses of 35–48 kDa (Figure 1) [22]. C1q is essential in the activation of CP, which is initiated upon recognition of antigen-aggregated immunoglobulins [23] or activated by *Klebsiella pneumoniae* porins (OMPs, 36 kDa) in an antibody-independent manner [24]. Roumenina et al. [25] reported that LPS derived from *S*. Typhimurium interacts specifically with the globular C1q domain in a calcium-dependent manner. Research carried out even earlier showed that C1q was bound more tightly to the re-form (core-defective mutant) of LPS than to the S form in *Salmonella* Minnesota [26].

In recent years, there has been a significant increase in the number of *Salmonella* strains resistant to antibiotics especially fluoroquinolones, which are the antibiotics of choice in intestinal infections [27]. *Salmonella* strains resistant to multiple drugs, such as third-generation cephalosporins and azithromycin, fluoroquinolones are also common worldwide [28]. QAS are commonly used disinfectants in many fields of medicine, industry, and agriculture. These compounds are membrane-active agents, and their action leads to the leakage of intracellular constituents, resulting in cytolysis. Interestingly, there is a scientific explanation of the linkage between an improper usage of biocides (e.g., in subthreshold levels) and the formation of microbial drug resistance [29,30]. The common mechanisms of cross-resistance to antibiotics and QAS may be the over-expression of genes encoding efflux pumps that extrude a wide range of compounds outside of the cells or the down-regulation of membrane porin channels (Figure 1) [31].

*Salmonellae* can multiply in the temperature range of 8–45 °C, but the most effective growth is observed at 37 °C. In terms of laboratory diagnostics, rods cultured on non-selective and non-differential media produce colonies with an even appearance, buttery texture, entire edges, and grayish white. Colonial morphology deserves attention, as it may reflect the differential expression of components on surfaces of bacterial cells within the colonies or suggest of weakened virulence [32,33]. The ability of microorganisms to attach to host surfaces is a critical stage for the initiation of surface colonization. Pathogenic factors involved in bacterial adhesion to host cells are hydrophobic surface, the ability to move, and synthesis of adhesins [34,35]. Among them, flagella and fimbriae have been found to play a vital role in the production of biofilms, particularly in the early phases when microcolonies are forming [36]. Effective adhesion results in biofilm formation. Bacterial biofilms constitute an important concern for the food industry. Within these structures, bacterial cells are sheltered against different unfavorable factors such as disinfectants, ultraviolet light radiation, osmotic changes, pH variability, dehydration, antimicrobial agents, host immune responses, and metal toxicity [37]. Microorganisms growing in biofilms are persistent to antibacterial treatment, requiring higher than clinical doses of antibiotics to eradicate [38].

In the present work, we have studied the phenotype connected to the virulence of *Salmonella* strains that were subjected to consecutive passages in 50% human serum from platelet-poor plasma (SPPP), sterile-filtered. We have also determined membrane protein (MPs) patterns (isolation of proteins based on Triton X-114) for each strain and examined the deposition of C1q and C3c molecules on the bacterial cells. In reference to our previous publication [39] serovar Isaszeg, in this paper, was considered as a control.

Serial exposure to SPPP was intended to create stressful conditions for bacteria, allowing us to observe numerous phenotypic changes. The obtained results have led to the following conclusions: (i) eight passages in SPPP may not be enough for the bacteria to become SR (*S*. Typhimurium ATCC 14028 P8, *S*. Senftenberg P8); (ii) C1q and C3c molecules were bound to *S*. Hammonia P8 and *S*. Typhimurium ATCC 14028 P8′s MPs, which composition changed after passaging; (iii) bacteria being exposed to sera may acquire MDR phenotype (*S*. Hammonia P8) and tolerance to QAS (Isaszeg P8 and *S*. Erlangen P8); (iv) passages in SPPP generated genetic changes within gene *fljB,* which translated to cells’ motility (*S*. Erlangen P8, *S*. Typhimurium ATCC 14028 P8); (v) the effect of temperature is more important in biofilm formation by *Salmonellae* than multiple passages in SPPP (*S*. Erlangen P8, *S*. Typhimurium ATCC 14028 P8, *S*. Enteritidis ATCC 13076, *S*. Senftenberg P8). 

## 2. Results

### 2.1. Confirmation of Salmonella Enterica with Slide Agglutination Test

Slide agglutination test was performed with polyvalent rabbit anti-*Salmonellae* serum to prove the identification of strains. Agglutination of the suspensions with antiserum was examined for twelve tested strains and observed with the naked eye. All the strains included in this paper (WT and P8) gave positive agglutination reactivity.

### 2.2. Serum Susceptibility Assay

Bactericidal activity of 50% SPPP at 37 °C was performed on six *Salmonella enterica* strains. The average number of colony-forming units (CFU/mL^−1^) was calculated from the colonies grown on the LB plates from bacterial suspensions. Because C components C2 and factor B are irreversibly denatured by mild heating, heat-inactivated SPPP (HISPPP) served as a negative control for complement-mediated lysis of serum. Results are presented as a ratio of surviving CFU in fresh SPPP vs HISPPP (Table 1). In HISPPP, bacteria survived in the range of about 100–122%. In 50% SPPP with active C, the bacterial survival ratio decreased to not more than 0.1% for *S*. Typhimurium ATCC 14028, *S*. Enteritidis ATCC 13076, and *S*. Senftenberg. Data published in 2017 show that serovars Isaszeg, Erlangen, and Hammonia were resistant to NHS after nine passages in NHS [16] and those variants after passages in NHS have been stored at −70 °C banked). After thawing and rejuvenating in 2021 strains showed sensitivity to SPPP at the levels of survival of 22.7% (*S*. Isaszeg), 8% (*S*. Erlangen), and (*S*. Hammonia) 23.3% (current experiments). Repeated experiments gave similar results. 

### 2.3. Passages of Bacteria in SPPP (Human Serum from Platelet-Poor Plasma)

After confirming sensitivity of tested strains to 50% SPPP we performed weekly passages of six *Salmonella enterica* WT strains. Before each serial passage, the number of cells had been adjusted to about 5.0 × 10^3^ when incubated in 50% SPPP or HISPPP as a control. The number of viable *Salmonellae* (CFU/mL^−1^) was determined by serial dilution on TSA agar at the beginning of the experiment (time 0) and after 180 min of incubation at 37 °C. All strains when incubated for 3 h in HISPPP (control) proliferated very intensively as the heating damaged the bactericidal activity of serum (Figure 1). With this assay, we showed that *S*. Isaszeg, *S*. Erlangen, *S*. Hammonia, and *S*. Enteritidis ATCC 13076 developed again (in comparison to results published in 2017 [16]) resistance to the bactericidal activity of serum after eight passages in 50% SPPP. Log_10_ in CFU/mL^−1^ increased from 3.5 to 7.0. These four strains have been adapted consequently to the presence of SPPP during a series of passages and at the end of the eighth passages they reached CFU values of 3.2 × 10^6^ ± 6.5 × 10^5^, 1.9 × 10^6^ ± 4.1 × 10^5^, 1.8 × 10^6^ ± 3.3 × 10^5^, and 2.3 × 10^6^ ± 8.1 × 10^5^, respectively from 7.0 × 10^3^ ± 3.2 × 10^3^, 3.3 × 10^3^ ± 1.2 × 10^3^, 2.0 × 10^3^ ± 8.2 × 10^2^ after the first passage. The survival ratio of the other two strains: *S*. Typhimurium ATCC 14028 and *S*. Senftenberg decreased to log_10_ = 1.5 or remained at a similar level of log_10_ = 3, respective, as evidenced also by the R-squared regression model. When it comes to bacterial percentage, the survival relative to the initial number of cells for four adaptants: *S*. Isaszeg P8, *S*. Erlangen P8, *S*. Hammonia P8, and *S*. Enteritidis ATCC 13076 P8 the values 228%, 158%, 222%, 177% were noted. *S*. Typhimurium ATCC 14028 and *S*. Senftenberg at the same time presented susceptibility to SPPP at the levels of 0.001% and 0.1%, respectively. 

### 2.4. Binding of C1q and C3c Complement Components to Salmonella enterica Cells

Complement component C3 is the central component of the C cascade and C1q is essential in CP activation. It is believed that the CP, not the AP, is necessary for the optimal lysis of complement-susceptible strains at both low and high serum concentrations. Here, we show the results of the deposition of C3c fragments from C3 decay and C1q protein on each of the four serum-resistant (SR) strains in relation to WT serum-sensitive (SS) strains. The tested bacteria were incubated in non-diluted SPPP for 12 min, and then the binding of C molecules was analyzed by indirect ELISA using appropriate primary and secondary antibodies. Detection of the signal was performed with *o*-phenylenediamine dihydrochloride and measured at 490 nm. Deposition of C1q and C3c occurred quite rapidly, because 12 min was enough significant differences (*p* < 0.05) to be noticed. The pattern of C1q deposition on the strains (Figure 2A) was more uniform in comparison to C3c binding (Figure 2B). In both C1q and C3c assays, the highest level of reaction was recorded for SS *S*. Isaszeg (not passaged). Significantly lower C1q deposition (*p* < 0.05) was demonstrated in five of six variants: *S*. Isaszeg P8, *S*. Erlangen P8, *S*. Enteritidis ATCC 13076 P8, *S*. Typhimurium ATCC 14028 P8, and *S*. Senftenberg P8. Moreover, it was observed that C3c molecules deposited less willingly only on two: *S*. Isaszeg P8, and *S*. Typhimurium ATCC 14028 P8. Together, these findings help us to formulate an initial hypothesis that the deposition of C1q and C3c on P8 *S*. *enterica* strains is somehow inversely related.

### 2.5. SDS-PAGE of Salmonella enterica Membrane Proteins (MPs)

Because many reports indicate the role of membrane protein (MPs) in SR phenomena, the experiments were aimed at developing MPs patterns (isolation based on detergent Triton X-114) for each tested strain and finding the differences between WT (SS) and P8 strains (SS or SR). Bacteria belonging to two working groups were incubated overnight at temperatures of 37 °C (Figure 3A) and 28 °C (Figure 3B) before the MPs isolation procedure was implemented. This step was taken on the basis of previous publications [40,41,42]. Prepared antigens were purified and then their quantitation was estimated with BCA assay. The amounts of proteins isolated from WT strains incubated at 37 °C was 6 mg/mL^−1^ and for those cultured at 28 °C 8 mg/mL^−1^. MPs concentration in P8 strains reached values of 4 mg/mL^−1^ at 37 °C and 3 mg/mL^−1^ at 28 °C. Next, for electrophoresis, the concentrations were adjusted to 12 μg/10 μL of proteins per well. The patterns obtained for the MPs with apparent molecular mass between about 25 and 75 kDa (selection of the molecular weight range according to [15,22,40] are shown in Figure 3. Many distinct protein bands were distinguishable in all paths; however, some of them were recognized as common to *Salmonella* strains, independently of incubation conditions. The group of typical *Salmonella* species bands contains about fourteen MPs with molar masses: 25, 30–32, 33–34, 36–37, 38–39, 42–43, 45–47, 55, 58–59, 62–63, 67–68, and 74–75 kDa. The detailed analysis has revealed a correlation between the various MPs band patterns and the susceptibility of the strains to the serum. The greatest diversity in the quantity of possessed MPs was observed for pairs *S*. Hammonia (SS)/*S*. Hammonia P8 (SR) and *S*. Typhimurium ATCC 14028 (SS)/*S*. Typhimurium ATCC 14028 P8 (SS). At 37 °C, *S*. Hammonia P8 overproduced five proteins with molar masses of 30, 36, 52, 60, and 65 kDa. At a temperature of 28 °C changes were even more distinct because of the presence of nine clearly visible bands: 36, 39, 42, 45, 52, 56, 60, 65, and 139 kDa. A reverse relationship was noticed for *S*. Typhimurium ATCC 14028 P8, which down-regulated MPs of molecular masses 54, and 61 kDa at both temperatures. 

### 2.6. Dot Blot C1q and C3c-Binding

After showing that C1q and C3c were efficiently bound to *S*. *enterica* cells, we determined the binding of C components to MPs isolated from bacteria preincubated at 37 °C. We expected to notice differences between the signal obtained for MPs acquired from SS (Hammonia, Typhimurium ATCC 14028, Typhimurium ATCC 14028 P8) and SR adaptant (Hammonia P8). Specimens of MPs 25-(0.24 mg/mL^−1^) (Figure 4A) and 50-diluted (0.12 mg/mL^−1^) (Figure 4B) in H_2_O_miliQ_ were loaded (1.5 μL) onto nitrocellulose prepared in advance. After being washed and non-specific binding blocked, MPs spots were treated with non-diluted SPPP for 20 min. at 37 °C. This time was sufficient for C activation and C components binding to bacterial antigens. To detect C1q binding to MPs and C3c bound neo-antigens, appropriate antibodies were used. The signal was detected with a TMB substrate. With this method, we were able to differentiate analyzed strains. As can be seen in Figure 4, C1q binding occurred on MPs coming from *S*. Hammonia, *S*. Hammonia P8, and *S*. Typhimurium ATCC 14028, wherein a stronger signal was obtained for MPs diluted 50×. In the case of C3c binding to MPs, no signal was detected, which was very similar to controls (K1, K2, K3). Results demonstrate effective C1q deposition on MPs (except of *S*. Typhimurium ATCC 14028 P8) and lack of the signal obtained for C3c.

### 2.7. Antibiotics and Biocides Susceptibility Profiling

Of the twelve *Salmonella enterica* strains tested with minimal inhibitory concentration (MIC) assay showed resistance to one or more antimicrobial substances. Drug resistance profiles specified on this basis are shown in Table 2. The group of nine strains was characterized by the resistance to trimethoprim-sulfamethoxazole, ciprofloxacin, and colistin and belonged to the so-called multidrug-resistant (MDR) strains. Three strains showed resistance to trimethoprim-sulfamethoxazole, and ciprofloxacin, while maintaining sensitivity to colistin. Among the studied strains, sensitivity to ampicillin, amoxicillin-clavulanic acid, and cefotaxime was found. The second group of antimicrobial compounds consisted of disinfectants containing QAS. MICs estimated for tested biocides were higher for *S*. Isaszeg P8 (SR) and *S*. Erlangen P8 (SR). Other values remained the same or slightly lower. The strains most resistant to antibiotics were the following: *S*. Erlangen (SS, SR), *S*. Hammonia P8 (SR), S. Typhimurium ATCC 14028 (SS), *S*. Enteritidis ATCC 13076 (SS, SR), and *S*. Senftenberg (SS). Elevated QAS MICs values were noted for S. Isaszeg (SR) and *S*. Erlangen (SR). In conclusion, passaging of *Salmonella* in sera could initiate resistance to colistin and to QAS, but it was dependent on the strain. 

### 2.8. Detection of Virulence Genes (VGs)

The results showing virulence-associated genes (VGs) are presented in Table 3 and Figure 5. *Salmonella* strains both wild-type (WT) and passaged (P8) in SPPP counterparts possessed in their genomes the following virulence genes: *invA*, *pagC*, *tolC*, *pgtE*, *fliC*, *OmpR,* and *csgD*. The only difference concerned the *fljB* gene, which was not detected in *S*. Erlangen (SS), *S*. Senftenberg (SS), *S*. Typhimurium ATCC 14028 P8 (SS), and *S*. Senftenberg P8 (SS). Considering the effect of passaging, the *fljB* gene was detected in the genome of *S*. Erlangen P8 and not detected in starting culture of *S*. Erlangen. The reverse happened in the case of *S*. Typhimurium ATCC 14028 P8, in which *fljB* was not detected. The study suggests that adaptation of *Salmonella enterica* to serum and overcoming immunological bactericidal mechanisms manifest in genetic changes in flagellar genes, especially *fljB*. 

### 2.9. Biofilm Formation by Salmonella enterica

The results showed that strains tested in this study produced biofilms on polystyrene microwell plates. The OD_570_ ranged from 0.108 to 0.556. A cutoff value of 0.011 (OD_c_) was used for strain classification according to the proposed [43] interpretation: OD ≤ OD_c_–non-biofilm producer; OD_c_ < OD ≤ 2 × OD_c_–weak-biofilm producer; 2 × OD_c_ < OD ≤ 4 × OD_c_–moderate-biofilm producer; 4 × OD_c_ < OD–strong-biofilm producer (Table 4). These results indicate that the mean OD_570_ of WT strains were similar to P8 serovars at 28 °C (strong-biofilm producers) with the exception of *S*. Enteritidis ATCC 13076 (SS). It has been recognized as a moderate-biofilm producer. The greater variability among serovars has been noticed at 37 °C. At this temperature, five strains showed the ability to form biofilms to a lesser extent as they were classified as moderate-biofilm producers and they were *S*. Erlangen (SS), *S*. Typhimurium ATCC 14028 P8 (SS), *S*. Enteritidis ATCC 13076 (SS, SR), and *S*. Senftenberg P8 (SS). 

### 2.10. Cell-Surface Hydrophobicity of Salmonella enterica

As shown in Table 5, strains that were basically strong hydrophobic (*S*. Isaszeg) or hydrophobic (*S*. Typhimurium ATCC 14028) became very strong hydrophobic after passages in 50% SPPP. The rest of the tested strains (*S*. Erlangen, *S*. Hammonia, *S*. Enteritidis ATCC 13076, and *S*. Senftenberg) were initially very strong hydrophobic and so they remained after eight passages in serum. Cell surfaces were classified as hydrophobic, strong hydrophobic and very strong hydrophobic because they aggregated in 1.2 M, 1.0 M, and 0.2 M ammonium sulfate, respectively. It seems that under stressful conditions (sera) bacteria are seeking to increase the hydrophobicity of cells’ surfaces. 

### 2.11. Colonial Morphology of Salmonella Strains Treated and Not Treated with SPPP

The streak plate technique across TSA agar plates was used to observe the morphology of isolated colonies of *Salmonella* strains passaged in SPPP or without prolonged exposure to sera. The assumption was that the passaging of strains may change somehow of colonial morphology. Results obtained for six representative isolates were recorded after 24 h and 48 h of incubation at 37 °C. Ultimately, the morphology of bacterial colonies did not change after passaging of strains in 50% SPPP. *Salmonella enterica* both WT and P8 variants produced on TSA medium typically white, smooth, with round edges colonies of similar size of 2–4 mm in diameter (Figure 6).

### 2.12. Motility of Salmonella enterica Passaged and Not Passaged in 50% of SPPP

In both the swarming and swimming assays, the soft agar plates were inoculated in the middle of the plate with 5 μL of cultures in log-phase of growth normalized to 1.0 OD (600 nm). Plates were incubated at temperatures of 28 °C and 37 °C for 12 and 18 h in triplicate. It was expected that previously noted changes within the *fljB* gene would reveal in *Salmonella* motility. On motility lab-agar containing 0.6% agar-agar supplemented with 0.5% glucose (swarming assay), bacteria did not spread around the spot (Figure 7). After 18 h, the spots remained the same in diameter and were more pronounced because of bacterial multiplication at the area of inoculation.

The expected differences emerged when 0.3% soft agar (0.3 g of agar-agar per 100 mL LB) was used (Figure 8). The addition of 0.05% TTC (2,3,5-triphenyl-2H-tetrazolium chloride), as a growth indicator, facilitated the observation of the edges of the growth. The first thing noticed was greater swimming motility of *Salmonella enterica* strains at 37 °C (*p* < 0.05). Regarding the comparison of WT and P8 strains, statistically significant differences (*p* < 0.05) occurred between *S*. Erlangen (SS) (48.3 mm ± 0.9 mm) and *S*. Erlangen P8 (SR) (56.0 mm ± 0 mm) at 28 °C and the same serovar at 37 °C, 70.3 mm ± 0.5 mm for WT and 90.0 ± 0 for P8 variant. The opposite differences in the diameters of bacterial cultures were noted for *S*. Hammonia (SS) (29.7 mm ± 1.2 mm) and *S*. Hammonia P8 (SR) (25.0 mm ± 0 mm) at 37 °C. Similar trends of less mobility for serum adaptants were seen for *S*. Typhimurium ATCC 14028 P8 (SS) (40.0 mm ± 0 mm) and *S*. Senftenberg P8 (SS) (39.0 ± 1.4 mm) (*p* < 0.05). No differences were observed for serovar Isaszeg (SS, SR) as well as Enteritidis ATCC 13076 (SS, SR). 

## 3. Discussion

*Salmonella enterica* is still an emerging pathogen [44] in some parts of the world. Research coming out from Malawi showed that bacteremia occurred without sepsis in children due to NTS *Salmonella* serotypes. It has been discovered that infections peaked during the rainy season and *S*. Enteritidis was the most frequently isolated serovar. In children aged 6 months or more, the presence of NTS was found to correlate with the incidence of malaria and with severe cases of anemia. *Salmonella* pathogens showed significant resistance to the antibiotics used which translated into high mortality in children. Also, relapses of NTS bacteremia that followed shortly after antibiotic therapy were particularly common in children infected with HIV [45]. Similar studies were conducted among children from Mozambique. These data confirmed the thesis that bacteremia with non-torment *Salmonella* serotypes was a significant problem in African countries [46]. Other studies by MacLennan et al. [47], also carried out on children from Malawi, showed that the presence of NTS-specific antibodies effectively reduced the number of bacteria in the blood. The lack of mentioned antibodies in sera was the cause of the bacteria survival, despite the proper functioning of the C system. More detailed analyzes revealed that all the tested strains had a long O-specific chain in LPS and the *rck* gene, the features thought to impart an effective defense against the C system. It was also found that disruption of LPS biosynthesis resulted in the destruction of bacteria by serum devoid of NTS-specific antibodies. In our research, *Salmonella enterica* strains were sensitive to non-immune SPPP (Table 1), despite possessing S-type LPS and containing NeuAc in O-Ag (sv. Erlangen, sv. Hammonia) [48]. Interestingly, sialic acid is thought to protect the host from autolytic attack by the AP [48]. Moreover, Goh et al. [49] pointed out an interesting issue, namely the concentration of exogenous C needed to sufficiently kill *Salmonella* may differ depending on the invasiveness of the isolate. While 20% baby rabbit serum was sufficient to kill the laboratory *S*. Typhimurium LT2 and *Salmonella* Paratyphi ACVD1901 strains, 75% BRS was needed to kill invasive African *S*. Typhimurium isolate D23580. 

Six *S*. *enterica*-resistant ciprofloxacin and trimethoprim-sulfamethoxazole have been serially passaged in SPPP to induce a series of phenotypic changes as a response/adaptation to unfavorable conditions. Obtained data indicate that one strain developed MDR and two of six P8 variants greater tolerance to biocides. *Salmonellae* more willingly demonstrated swimming motility at 37 °C than at 28 °C; however, they developed tolerance to the bactericidal activity of serum manifested in instability of the phase 2 flagellin *fljB* gene. Lower mobility at 28 °C promoted biofilm formation at 28 °C and passaging in 50% sera influenced greater hydrophobicity of *Salmonella* cells. 

In our study, non-immune SPPP was used to create stressful conditions for *Salmonella enterica* and make them adapt. Three of the six strains involved in the research had been previously (2017) sequentially passaged in NHS obtained from volunteers and became SR, which was related to the elongation of LPS O-Ag [16]. As it turned out, eight passages of SS isolates in 50% SPPP was successful to obtain four (*S*. Isaszeg, *S*. Erlangen, *S*. Hammonia, S. Enteritidis ATCC 13076) SR adaptants (Figure 1). These experiments confirmed the results obtained on NHS [16]. The plaque numbers of bacteria fluctuated after each serum treatment to finally approach (rising trend line) the abundance of cells cultivated in HISPPP. Some investigators pointed out that the modulation of protein expression (protein profiles) during in vitro passage caused changes in the virulence and immunogenicity phenotype of *Burkholderia pseudomallei* both reference strain and clinical isolates. This is consistent with the results by Somerville et al. [50] who found an increase in *Staphylococcus aureus* virulence after passaging and Mankoski et al., who showed that passaged *Helicobacter pylori* in gnotobiotic piglets exhibited increased expression of the *flaA* flagellin gene, as well as increased bacterial colonization [51]. 

In this work, we undertook research extended by membrane proteins (MPs) analysis (Figure 3) as we hypothesize that some classes of MPs may determine SR and other phenotypic features such as resistance to antibiotics, tolerance to disinfectants, or binding C components. At first, it has been shown that both WT *Salmonellae* and P8 variants bound C3c and C1q molecules, wherein less variation in the levels of activations was observed in the C1q assay. This may be explained by a huge group of common MPs in *Salmonellae* (Figure 3) that conditioned binding of C1q complement protein (Figure 4). Interestingly, the downregulation of most surface proteins in *S*. Typhimurium ATCC 14028 P8 leads to weaker binding of the C1q subcomponent. We found out that rapid C3c and C1q conversion on the surfaces of *Salmonella* cells. The time allowed for the reaction of serum with bacterial suspensions was optimized to 12 min. With the dot-blot technique, we have demonstrated that MPs were antigens recognized by C1q and C3c (Figure 4), which is in accordance with the work by Alberti et al. [52] and Futoma-Kołoch et al. [22]. Alberti et al. [52] provided data that major outer membrane proteins in *K*. *pneumoniae* bound C1q, while strains possessing LPS O-Ag bound less Clq. In turn, Futoma-Kołoch et al. [22] suggested that activation of C3 serum protein was dependent on the sialic acid contents in the LPS as well as on the presence of outer membrane proteins (OMP). In this paper, dot-blot was applied for MPs isolated and purified from four representative strains, SS *S*. Hammonia, SR *S*. Hammonia, SS *S*. Typhimurium ATCC 14028, and SS *S*. Typhimurium ATCC 14028 P8. C1q was detected on MPs from both SS *S*. Hammonia and SR *S*. Hammonia P8 as well as on antigens obtained from SS *S*. Typhimurium ATCC 14028. Considering the differences between MPs patterns, we suggest somehow involvement of MPs or OMPs with the tested C complement components interactions [22,39].

Due to the lack of studies on how the serial sera treatment affects susceptibility to antibacterials, the next goal was to analyze the MICs of clinically relevant classes of antibiotics of six WT *S*. *enterica* strains, and six P8 variants (Table 2). It turned out that twelve tested strains were resistant to ciprofloxacin and trimethoprim-sulfamethoxazole. Four WT strains: *S*. Erlangen, *S*. Typhimurium ATCC 14028, *S*. Enteritidis ATCC 13076, and *S*. Senftenberg, and five P8 adaptants: *S*. Erlangen, *S*. Hammonia, *S*. Typhimurium ATCC 14028, *S*. Enteritidis ATCC 13076 and *S*. Senftenberg were also resistant to colistin. Considering the passaging process, adaptation to SPPP translated into tolerance of *S*. Hammonia P8 to colistin and cefotaxime, *S*. Isaszeg P8 to ciprofloxacin, and *S*. Senftenberg to amoxicillin-clavulanic acid. The relationship between the adaptation to sera and susceptibility to biocides was also found. When it comes to the determination of MICs for QAS, *S*. Isaszeg P8 and *S*. Erlangen P8 showed higher tolerance. The most stable were two reference strains: *S*. Typhimurium ATCC 14028 P8 and *S*. Enteritidis ATCC 13076 P8, for which MICs of drugs and disinfectants did not change. 

Colistin has been extensively used for the control of enteric infections in farm animals for the prevention of infections in humans. The mechanism of action is solubilizing the bacterial cell membrane, resulting in a bactericidal effect [53]. Moreover, colistin-resistant bacteria also share resistance to other types of antibiotics used such as aminoglycosides, tetracycline, sulfonamide, and trimethoprim, lincosamide, b-lactams, quinolones, and third-generation cephalosporins [54]. Our data seem to support the concept of sharing resistance, as our strains were resistant to both ciprofloxacin and trimethoprim-sulfamethoxazole. Additionally, pathogens that got used to the presence of serum may develop the MDR phenotype, which is a dangerous phenomenon. In many Gram-negative bacteria such as *Klebsiella*, *Escherichia coli*, *Shigella*, *Citrobacter*, *Proteus*, *Enterobacter*, and *Salmonella*, the most common LPS modifications were associated with increased MICs to colistin. The efflux is also likely involved in colistin resistance, often resulting from combined resistance mechanisms of defects in OMPs and structural modification of the LPS [55]. Most of the tested in this work strains were resistant to colistin, except for *S*. Isaszeg, which kept sensitivity after prolonged treatment with SPPP. We suppose that truncated LPS, devoid of the O-specific part in *S*. Isaszeg may increase the effectiveness of colistin. Another isolate *S*. Hammonia P8 has become resistant to colistin as well as less sensitive to cefotaxime. In this case, the overproduction of MPs (Figure 3) might be the dominant mechanism of resistance. 

In subsequent stages of research, VGs prevalence analysis was carried out. In all tested strains (WT, P8), serum resistance genes *pagC* and *pgtE*, invasiveness genes *invA*, *tolC*, biofilm-forming genes *OmpR*, and *csgD* as well as fimbrial subunits gene *fliC* were detected. The only difference concerned the absence of *fljB* in *S*. Erlangen (SS), *S*. Senftenberg (SS), *S*. Typhimurium ATCC 14028 P8 (SS), and *S*. Senftenberg P8 (SS) (Table 3, Figure 5). Developed tolerance to the bactericidal activity of serum manifested in instability of phase 2 flagellin *fljB* gene. Intriguing reports came from Lyu et al. [56] who showed that *Salmonella* cells that do not express flagella are more tolerant to antibiotics. Arieta-Giasola et al. [57] indicated at least thirteen different types of *fljB* deletions in *Salmonella* 4, [5], 12:I:-, a monophasic variant of *S*. Typhimurium. It seems that genetic changes within the sequences encoding flagella in *Salmonella* (H antigen) may pose problems in correct identification with serological methods. 

It is likely that *S*. *enterica* has an unstable genome, especially in the region of flagellar genes, which could be somehow beneficial for bacterial adaptation. To check the functionality of this change, strains were tested against swarming motility (0.6% agar-agar). Surprisingly, there were no differences in the motility of all tested isolates (Figure 7); therefore, a swimming motility assay was performed (agar containing 0.3% agar-agar), which helped to observe significant differences (Figure 8). These data indicate that bacterial adaptation to serum is connected to changes in motility. 

Additionally, our aim was to compare colonial morphotypes produced by wild-type strains and those which adapted to serum. We were inspired by other studies which showed that the characteristics of a given strain may be extremely different after serial passages for many years on bacteriological media, so the number of cultures should be carefully defined for each experiment [58]. In our case, the acquisition of so many phenotypic features by *Salmonellae* passaged in sera did not influence the morphotypes of colonies. 

The biofilm-forming ability is dependent on numerous factors such as the growth phase of the cells, growth medium, contact time, properties of the inert material, and environmental parameters such as pH and temperature. The capabilities of biofilm formation coupled with motility and hydrophobicity may differ with respect to the species of *Salmonella* and their environment. This could be problematic in the food industry, since biofilms protect the bacteria from antibiotics, sanitizers as well as other environmental factors [38,59]. In addition, their survival strategies based on forming biofilms make them incredibly versatile and adept pathogens [37]. Hydrophobic interaction has been thought to be a major event in the attachment of bacteria to host cells or even the adherence of bacteria to dental surfaces [35,60]. To investigate hydrophobicity in *S*. *enterica* we performed the SAT test, which is a widely used simple method, earlier optimized for *Staphylococcus* and *Salmonella* species [61]. We determined that the hydrophobicity increased in P8 strains, from strong hydrophobicity in *S*. Isaszeg to very strong hydrophobicity in *S*. Isaszeg P8, and from hydrophobicity in *S*. Typhimurium ATCC 14028 to very strong hydrophobicity in *S*. Typhimurium ATCC 14028 P8. The remaining strains had very strong hydrophobicity. Similarly, in other studies, the cell surface hydrophobicity, and additionally haemolysin and mannose-resistant phenotypes were significantly higher among sepsis *E. coli* isolates as compared to the control group (*p* < 0.01) [37]. As other sources show, *Salmonella* flagella may be involved in early adhesion and microcolony development [62]. In our work, serial passages in SPPP caused genetic changes in the *fljB* gene in two strains: *S*. Erlangen P8 and *S*. Typhimurium ATCC 14028 P8, which was manifested in swimming motility but not in the intensity of biofilm production. The only mutation in the *fljB* gene seemed to be insufficient to disturb biofilm production. Moreover, *Salmonella* species in this study revealed biofilm potential at different temperature regimens: at 28 °C and 37 °C. The foremost observation revealed stronger biofilm formation at 28 °C than at 37 °C. This shows that *S. enterica* has developed multiple adaptation strategies to stress conditions in concomitance with previous literature [63]. Using the tube biofilm assay (TBA) to research biofilm development on cholesterol by *Salmonella* Typhi, other researchers have discovered that the flagellin subunit (FliC) is essential for early cholesterol-coated surface attachment, and the loss of outer-membrane protein C (OmpC) impeded cholesterol binding and biofilm formation [64].

## 4. Materials and Methods

### 4.1. Bacterial Strains

The study was carried out on three strains of *Salmonella* O48 serogroup characterized by the presence of sialic acid in lipopolysaccharides: *Salmonella enterica* subsp. *enterica* serovar Isaszeg, *Salmonella enterica* subsp. *salamae* sv. Erlangen, and *Salmonella enterica* subsp. *salamae* sv. Hammonia. These strains were obtained from the Polish Collection of Microorganisms (PCM), Hirszfeld Institute of Immunology and Experimental Therapy, Polish Academy of Sciences, Wroclaw, Poland. Another strain *Salmonella enterica* subsp. *enterica* sv. Senftenberg was isolated from poultry food samples in 2014 at the LAB-VET Veterinary Diagnostic Laboratory (Tarnowo Podgorne, Poland) by the procedures approved by Polish Centre for Accreditation. It was serotyped in the National Serotype *Salmonella* Centre (Gdansk, Poland). Laboratory protocols were also applied for reference strains: *S*. *enterica* subsp. *enterica* sv. Typhimurium ATCC 14028 and *S*. *enterica* subsp. *enterica* sv. Enteritidis ATCC 13076. The list of strains used in this work is presented in Table 6. 

### 4.2. Sera

Human serum from platelet-poor plasma (SPPP), sterile-filtered, mycoplasma, and virus tested was purchased from Sigma-Aldrich (Poznan, Poland). SPPP was frozen in 1.0-mL aliquots and stored at −70 °C for no longer than six months. The serum was thawed immediately before each usage and the portion was used only once. In passages and ELISA tests, 50% SPPP was used and in dot blot assay 100% SPPP. Sera decomplemented by heating at 56 °C for 30 min (HISPPP, heat-inactivated serum) was used as a control.

### 4.3. Antibiotics and Disinfectants

Antibiotics: ampicillin (AMP), amoxicillin/clavulanic acid (AMC), colistin (COL), cefotaxime (CTX), ciprofloxacin (CIP), and trimethoprim-sulfamethoxazole (TMP-SMX) (BioMérieux, Marcy l’Etoile, France). Disinfectants: commercial biocides: Virusolve+^®^ (Amity International, Barnsley, UK); Desprej^®^ (Biochemie, Bohumin, Czech Republic); benzalkonium chloride, BAC (Pol-Aura, Roznowo, Poland). These disinfectants were selected as representatives of biocides containing QAS.

### 4.4. Salmonella Serum Bactericidal Assay

Samples of bacteria cultured in Luria Bertani (LB, Biomaxima, Lublin, Poland) liquid medium were assayed for survival against the killing effect of 50% SPPP (Sigma-Aldrich, Poznan, Poland) at 37 °C as described previously [49], with minor modifications. Experiments were carried out in triplicate. Regarding the method, 5 μL viable *Salmonellae* at 3 h log-growth phase was added to 45 μL PBS-diluted serum (final *Salmonella* concentration in approximately 1.5 × 10^6^ CFU/mL^−1^) and incubated at 37 °C. For the control, HISPPP was performed by incubating the sera at 56 °C for 30 min. The number of viable *Salmonellae* (CFU/mL^−1^) was determined on LB agar after 0 and 180 min of incubation. Experiments were performed in triplicates and repeated three times. 

### 4.5. Prolonged Exposition (Passages) of Bacteria in SPPP (Human Serum from Platelet-Poor Plasma)

*Salmonella enterica* strains were taken from preserved glycerol stocks (−70 °C) and revived in tryptic soy broth (TSB, Biomaxima, Lublin, Poland) and streaked on MacConkey Agar (Biomaxima, Lublin, Poland) plates. Well-isolated clear colonies from MacConkey Agar plates were picked and further cultured on tryptic soy agar (TSA, Biomaxima, Lublin, Poland). Before starting the passaging assay, a serum bactericidal test was performed for each strain. Serovars Typhimurium ATCC 14028, Enteritidis ATCC 13076, and Senftenberg were totally sensitive to 50% SPPP, with survival below 0.1% (Table 1). Three more serovars Isaszeg, Erlangen, and Hammonia characterized as resistant to NHS in 2017 [16] showed sensitivity to 50% SPPP [NO_PRINTED_FORM] At the day of each passaging incident, colonies from TSA agar were transferred to TSB to reach a log-growth phase of the bacterial population. Bacterial suspensions (3.0–5.0 × 10^3^ or 3.0–5.0 × 10^5^) were mixed with just thawed sera (Sigma-Aldrich, Poznan, Poland) or HISPPP sera in the proportion of 1:1 (50%). The number of viable *Salmonellae* (CFU/mL^−1^) was determined by serial dilution on TSA agar at the beginning of the experiment and after 180 min of incubation at 37 °C. CFU data were converted to log values. Passages were independent, time-consuming, spread over time (eight weeks) experiments. Each tested strain was passaged in SPPP eight times and confirmed three times. The number of passages was estimated in a previous study [16] as sufficient for the adaptation of bacteria to diluted sera. Bacterial cells obtained from the eighth passages were stored in Microbank^®^ vials (Biomaxima, Lublin, Poland) under −70 °C for further analysis. 

### 4.6. Slide Agglutination Assay

Serological identification of *Salmonellae* taken from preserved glycerol stocks (−70 °C) and revived in TSA agar (Biomaxima, Lublin, Poland) was performed with SIT agglutination kit (Immunolab, Gdansk, Poland). Bacteria were cultured on TSA agar at 37 °C for 20 h. One drop of NaCl (control) and polyvalent serum (rabbit anti-*Salmonellae*) were placed on degreased glass slides. Bacterial colonies were picked from the agar plates and put near previously prepared drops. Next, the bacteria were rubbed with drops to obtain a homogenous suspension. The results were noted after 30-s of swaying movement.

### 4.7. Enzyme-Linked Immunosorbent Assay (ELISA)

Complement C1q binding assay for bacterial cells was carried out by indirect ELISA according to Alberti et al. [52] with minor modifications. Microtiter plate wells (BRANDplates^®^, pureGrade, Wertheim, Germany) were coated with bacterial suspensions. The coating was achieved by dispensing 50 μL (10^7^ CFU/mL^−1^) of the bacterial suspensions (*Salmonellae* incubated) and drying it overnight at RT. On the next day, the wells were washed two times with 100 μL/well of PBST (0.05% Tween 80 in PBS) and twice with PBS. The remaining drops were removed by patting the plate after the final wash. After that, 100 μL of 5% milk was added to block non-specific binding. The plates were covered with adhesive plastic films and incubated for 1.5 h at 37 °C. Next, plates were emptied and washed as above and filled with 50% SPPP (Sigma-Aldrich, Poznan, Poland) diluted in PBS and left for 12 min. Then, the wells were washed twice with 100 μL/well of PBST (0.05% Tween 80 in PBS) and twice with PBS. The remaining drops were removed by patting the plate after the final wash. The plates were incubated with mouse anti-human C1q monoclonal antibody (1:1000 dissolved in 1% low-fat milk) (GeneTex, Irvine, CA, USA, 1.0 mg/mL^−1^), then with donkey anti-mouse IgG antibody (1:1000 dissolved in 1% low-fat milk)/HRP (horseradish peroxidase-conjugated, GeneTex, 1.0 mg/mL^−1^) and finally with *o*-phenylenediamine dihydrochloride (Thermo Fisher Scientific, Waltham, MA, USA) to develop alkaline phosphatase and measured at 490 nm. A complement C3c binding assay for bacterial cells was carried out by the same method. All the steps were very similar to those above, but another set of antibodies was used: polyclonal rabbit anti-human C3c (DakoCytomation, Glostrup, Denmark) (3.3 g/L^−1^) as the primary antibody and polyclonal goat anti-rabbit immunoglobulins/HRP (horseradish peroxidase-conjugated, DakoCytomation, Glostrup, Denmark) (0.05 mol/L^−1^) as a secondary antibody. The plates were also developed with *o*-phenylenediamine dihydrochloride (Thermo Fisher Scientific, Waltham, MA, USA) and measured at 490 nm. Control wells contained PBS instead of bacteria, or lack of primary antibodies. 

### 4.8. Membrane Proteins (MPs) Isolation and Preparation 

The isolation of MPs from twelve strains was performed with ReadyPrep^TM^ Protein Extraction Kit, Membrane I (Bio-Rad, CA, USA). A set of chemicals used temperature-dependent phase partitioning and Triton X-114 detergent to prepare fractions that are enriched in membrane and hydrophobic proteins. MPs were isolated from WT strains and *Salmonellae* cells obtained after the eighth exposition to 50% SPPP. Protein isolations were preceded by the overnight cultivation of P8 strains at both 28 °C and 37 °C. Total MPs concentration was measured using a BCA Protein Assay Kit (Pierce^®^ Thermo Fisher Scientific, Waltham, MA, USA). MPs samples were purified with a ReadyPrep^TM^ 2-D cleanup kit (Bio-Rad, CA, USA). After the isolation, the final pellet was resuspended immediately in 4× Laemmli Sample Buffer (Bio-Rad, CA, USA) diluted in deionized water and solubilized at 100 °C for 5 min. 

### 4.9. Polyacrylamide Gel Electrophoresis (SDS-PAGE) of MPs

MPs were analyzed according to the Laemmli [65] buffer system using 5.0% stacking gel and 12.5% separating gel. Ten-microliter samples (12 μg) were applied to the wells. Five microliters of pre-stained protein marker (10–245 kDa, AppliChem, Darmstadt, Germany) were loaded in each gel run. Electrophoresis was performed at RT using a constant current of 19 mA in 1× solution of Tris/Glycine/SDS running buffer (Bio-Rad, CA, USA). Separated protein bands were detected with a silver staining kit (TH.GEYER, Warszawa, Poland). Results were confirmed in three independent experiments. 

### 4.10. Molecular Analyses of MPs Patterns

The gels were captured with the Quantity One^®^ 1-D Analysis Software, v. 4.6.6. and analyzed with Image Lab. V. 4.1 (Bio-Rad, CA, USA). 

### 4.11. Dot Blot Protein-Binding Assay 

Having PVDF nitrocellulose membranes (Bio-Rad, Hercules, CA, USA) ready, a point was marked by pencil to indicate the region is going to be blot. Using narrow-mouth pipette tips, 1.5 μL of diluted MPs were spotted at the center of pencil points. To minimize the area of solution penetration, specimens were applied slowly. Then, the membranes let dry. Next, non-specific sites were blocked by soaking in PBS containing 5% low-fat milk for 1 h (RT), washed twice with PBST (0.05% Tween 80 in PBS), twice with PBS, and let dry. In the next step, 1.5 μL of non-diluted SPPP (Sigma-Aldrich, Poznan, Poland) were spotted and membranes were incubated at 37 °C for 20 min. After that time, the membranes were washed as before and let dry. To detect bound C1q complement proteins, primary antibody–mouse anti-human C1q monoclonal antibody (1.0 mg/mL^−1^), 1:1000 dissolved in 1% low-fat milk) (GeneTex, Alton, CA, USA) and secondary antibody donkey anti-mouse IgG antibody (1.0 mg/mL^−1^), 1:1000 dissolved in 1% low-fat milk) (GeneTex, Alton, CA, USA) conjugated with HRP were used and incubated for 30 min at RT. To detect bound C3c polyclonal rabbit anti-human C3c (DakoCytomation, Glostrup, Denmark; 3.3 g/L^−1^) as primary antibody and polyclonal goat anti-rabbit immunoglobulins/HRP (horseradish peroxidase-conjugated, DakoCytomation, Glostrup, Denmark; 0.05 mol/L^−1^) as a secondary antibody was used. Between and after incubations with antibodies the membranes were washed and dried as described above. Detection of signal (blue) was performed with a TMB substrate kit for peroxidase (Vector laboratories, Newark, CA, USA). Controls for binding in the dot blot are defined in the description of Figure 4. Since SS *S*. Typhimurium ATCC 14028 was giving one of the highest signals in ELISA assays (Figure 2), it was regarded as a positive control.

### 4.12. Antimicrobial Susceptibility Testing

The minimal inhibitory concentrations (MICs) of antibiotics (AMP, AMC, COL, CTX, CIP, TMP-SMX) were determined by the method of successive dilutions of antibiotics in Mueller-Hinton solid medium (Sigma-Aldrich, Poznan, Poland) in accordance with CLSI recommendations [66]. The lowest inhibitory concentration (MIC) of their growth was determined. For the test control, the reference strain was used *E. coli* ATCC 25922. The cultures were incubated for 18 h at 37 °C. The results were interpreted in accordance with EUCAST recommendations [67]. Biocide (Virusolve+^®^, Desprej^®^, BAC) susceptibility testing (BST) was performed in accordance with the standardized protocol by Schug et al. [68]. In detail, biocide dilutions were prepared in sterile demineralized water. Two-fold dilution series were used (100%, 50%, 25%, 12.5%, 6.25%, 3.13%, 1.56%). All wells of the microtiter plate were inoculated with 100 μL of the bacterial suspensions (3 × 10^8^ CFU/mL^−1^), resulting in a final volume of 200 μL/well and a concentration of 1 × TSB. Growth controls were included by adding the 100 μL inoculum to row H of the microtiter plates. In addition, sterility controls for media were prepared in separate tubes. The MIC was defined as the lowest concentration that does not yield visible bacterial growth after 24 h. MICs for antibiotics and BST assay were performed before the first passage of *Salmonellae* in 50% SPPP and after the last, that was the eighth passage in sera. 

### 4.13. Detection of Selected Virulence Genes (VGs)

Polymerase chain reactions (PCR) were performed to detect invasiveness genes (*invA*, *tolC*), serum resistance genes (*pagC, pgtE*), fimbrial subunits genes (*fliC*, *fljB*), and biofilm forming genes (*OmpR*, *csgD*) [69,70,71,72]. Primers and functions of the targeted genes are shown in Table 7. Primer pairs were synthesized by Genomed (Warszawa, Poland), rehydrated in 10 mM Tris-HCl (pH 8), and stored at –20 °C until used. Each reaction mixture consisted of 12.5 µL PCR Mix Plus Green (0.1 U/µL DNA Taq Polymerase, 4 mM MgCl_2_, 0.5 mM of dNTPs (A&A Biotechnology, Gdansk, Poland); 0.125 µL of each primer and 2 µL of DNA template. The mixture was adjusted with sterile, nuclease-free water (A&A Biotechnology, Gdansk, Poland) to give 25 µL. The reactions were performed in conditions as follows: initial denaturation (94 °C, 5 min), 32 cycles of denaturation (94 °C, 1 min), annealing (65 °C, 45 s), extension (72 °C, 1 min), and final extension (72 °C, 10 min) with PCR Thermal Cycler T100 (Bio-Rad, CA, USA). PCR products were separated in 2% agarose gel with Midori Green Advance DNA Stain (Nippon Genetics, Duren, Germany). Electrophoresis was conducted for approximately 60 min at a voltage of 100 V. DNA products were detected with a Gel Doc camera system (Bio-Rad, CA, USA) and analyzed with Quantity One software (Bio-Rad, CA, USA). DNA mass markers (range: 100–1000 bp) and (range: 100–3000 bp, A&A Biotechnology, Gdansk, Poland) were used to estimate PCR product sizes.

### 4.14. Biofilm Formation

The biofilm formation was evaluated in 96-well polystyrene U-bottom microplates according to the method by O’Toole and Kolter [74] with minor modifications by Lamas et al. [43]. The aim of the experiment was to compare the intensity of biofilm formation by wild-type strains and their passaged counterparts. Wells of the microplates were filled with 100 μL of bacterial suspension (OD 0.1) prepared in TSB. Then, plates were incubated aerobically for 24 h and 48 h at temperatures of 28 °C and 37 °C. Each isolate was tested in triplicate. After incubation, the wells were emptied and washed twice with 200 μL of distilled water to remove free-floating cells. The bacteria that were attached to the walls were then fixed by adding 200 μL of 96% methanol for 15 min. The plates were emptied and air dried, and the wells were stained with 200 μL of 0.1% crystal violet solution (Sigma-Aldrich, Poznan, Poland) for 5 min. The excess crystal violet was depleted and washed three times with 200 μL of distilled water. The microplates were air-dried, and the dye that was bound to the adherent cells was resolubilized with 200 μL per well of 33% glacial acetic acid. The optical density (OD) of each well was measured at 570 nm with a plate reader (ASYS UVM 340, Biochrom, Cambridge, UK). The cutoff OD (ODc) was defined as three standard deviations above the mean OD of the negative controls. Thus, isolates were classified as a nonbiofilm producer (OD ≤ ODc), a weak biofilm producer ODc < OD ≤ (2 × ODc), a moderate biofilm producer (2 × ODc) < OD ≤ (4 × ODc), and a strong biofilm producer (4 × ODc) < OD.

### 4.15. Cell-Surface Hydrophobicity

Cell-surface hydrophobicity was measured by the salt aggregation test (SAT). In this assay, surface hydrophobicity is inversely correlated with the salt concentration that is required to mediate the agglutination of bacteria. SAT was performed by slide agglutination of bacteria with varying concentrations of ammonium sulfate, as described [75]. Bacterial suspensions (2.7 × 10^9^ CFU/mL^−1^) in PBS, pH 7.4 were mixed with ammonium sulphate (Alchem, Torun, Poland) solutions at final molar concentrations of 2.0, 1.8, 1.6, 1.4, 1.2, 1.0, 0.4, 0.2, 0.06 and 0.02. A 0.01-mL aliquot of each bacterial suspension was mixed on a glass microscope slide with an equal volume of ammonium sulfate at each concentration. The slides were gently rotated by hand and observed visually after 2 min for the presence of bacterial aggregation. The lowest concentration of ammonium sulphate at which bacteria aggregated was determined. Based on the SAT values, the strains were classified as: 0.02–0.2 mol/L^−1^, very strongly hydrophobic; 0.4–1.0 mol/L^−1^, strongly hydrophobic; 1.2–1.6 mol/L^−1^, hydrophobic; and 1.8–2.0 mol/L^−1^, hydrophilic.

### 4.16. Colonial Morphology of Salmonella Strains

The streak plate technique was used to compare colonies produced by WT and P8 strains. The inocula were streaked across TSA (Biomaxima, Lublin, Poland) and plates were incubated at 37 °C for 24 h. Well-isolated colonies were observed visually (Figure 6).

### 4.17. Motility Assay

Swimming and swarming motility were tested according to Ray et al. [76] method with modifications. All tested *Salmonella* strains (WT and P8) were cultured overnight on solid nutrient agar (Biomaxima, Lublin, Poland) to obtain single colonies. The soft agar plates were then inoculated with 5 μL of cultures normalized to 1.0 OD (600 nm), in the middle of the plate. Plates were incubated at temperatures of 28 °C and 37 °C for 18 h in triplicate. For the swarming assay (Figure 7), 20 mL of freshly made motility lab-agar (agar 0.6%, Biomaxima, Lublin, Poland) with 0.5% glucose were poured for Petri dishes. After drying at RT, the plates were inoculated with 5 μL 0.1 OD (600 nm) of respective culture in the logarithmic phase of growth. The plates were incubated at 28 °C and images were taken after 12 and 18 h. For the swimming motility assay (Figure 8), freshly made 0.3% soft agar (0.3 g of agar-agar per 100 mL LB) with the addition of 0.05% TTC (2,3,5-triphenyl-2H-tetrazolium chloride, Merck, Darmstadt, Germany) was used. Then 20 mL of soft agar was poured into Petri dishes and dried at room temperature (RT) under sterile conditions. The swimming range of the bacteria was assessed visually as a red zone due to the indicator used in the test. The diameter (mM) of a red zone was measured and the mean of three replicates was calculated. 

### 4.18. Statistical Analysis

The results of all the experiments are given as a mean value ± SD of 3 independent experiments. Comparisons between the binding of C3c and C1q to bacterial cells in the ELISA assay were made with the ANOVA Friedman and Kendall rank correlation coefficient test. The differences in swimming motility were analyzed with a parametric t-test for independent samples using Statistica v. 13 software (StatSoft, Krakow, Poland). Differences between groups were considered statistically significant for *p* values < 0.05. 

## 5. Conclusions

In summary, this study showed that growing *S*. *enterica* in a stressful environment, i.e., serum, brings numerous phenotypic changes connected with their virulence properties. Depending on the strain, bacteria may develop greater tolerance to serum (Figure 1A–D), antibiotics, and biocides (Table 2) [18] switch motility and hydrophobicity (Figure 8, Table 5) showing for the first time in this paper Some of these features may correlate with the content of surface protein antigens (MPs) (Figure 3) [18] that are a kind of fingerprint of the process of adaptation. In our opinion, interpretation of any data obtained for *Salmonella* species and probably other Gram-negative bacteria might consider different temperature ranges of cultivation. For example, swimming motility (Figure 8) and formation of bacterial consortia (Table 4) were more efficient at 28 °C. The influence of temperature is visible in MPs patterns (Figure 3), presumably determining bacterial pathogenicity. It has been shown previously that the presence of S-type LPS and its elongation protected *Salmonella* strains from the bactericidal action of NHS [16]. This work shows that MPs (isolated with Triton X-114) may be also involved in this process through interactions with C1q and C3c subcomponents (Figure 4) [52]. Despite phenotypic diversity occurring after prolonged exposition to SPPP, morphotypes of colonies remained unchanged on basic media (Figure 6) which in everyday practice may be confusing. *Salmonella* stable morphotypes distinguished by altered virulence may lull vigilance during laboratory work with life-threatening strains [33]. 

## Data Availability

The data presented in this study are available on request from the corresponding authors.

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
