# Peer review of "The Prolonged Treatment of Salmonella enterica Strains with Human Serum Effects in Phenotype Related to Virulence"

_ijms, 2023, doi:10.3390/ijms24010883_

Round 1

Reviewer 1 Report

The study by Futoma-Koloch et al investigates the effect of prolonged treatment of various Salmonella strains with human serum on the phenotype and virulence related genes. The study is well carried out and presents interesting observations. The results are properly presented and well described and discussed in the context of existing literature. Some of my minor suggestions are listed below:

1. Line 232: The rationale for use of two different temperature conditions should be justified and a reference should be cited.

2. Line 239: The rationale for selection of the molecular weight range should be justified and a reference should be cited.

3. It is unclear what the bottom half of figure 3 represents. For figure 2 - major tick marks for y axes are missing. 

4. For figure 4 - A higher exposure blot will look more convincing. Please clarify the reason for the appearance of the box in the K3 area. Also clarify if there are positive and negative controls for binding in the dot blot. If not kindly comment or provide the necessary controls.

5. For Table 3 and Figure 5 please explain why gene levels are looked at. What happens to the gene expression levels of the virulence associated genes? Please comment why a qPCR should or should not be performed. 

6. References to previous literature or results presented in the study should be provided for the statements made in lines - 530, 533, 816-818, 819-820.

7. Discussion section title is missing. The manuscript needs extensive English editing to correct for grammatical errors.

Author Response

Thank you for valuable engagement in reviewing the manuscript. Below we have listed answers to your suggestions.

Line 232: The rationale for use of two different temperature conditions should be justified and a reference should be cited.

We agree with that. There are many papers in which the level of expression of OMPs or their electrophoretic patterns at different temperatures have been investigated. We have decided to cite some of them:

Reeks BY, Champlin FR, Paulsen DB, Scruggs DW, Lawrence ML. Effects of sub-minimum inhibitory concentration antibiotic levels and temperature on growth kinetics and outer membrane protein expression in Mannheimia haemolytica and Haemophilus somnus. Can J Vet Res. 2005 Jan;69(1):1-10. PMID: 15745216; PMCID: PMC1142163.

D E, Orange N, Saint N, Gurillon J, De Mot R, Molle G. Growth temperature dependence of channel size of the major outer-membrane protein (OprF) in psychrotrophic Pseudomonas fluorescens strains. Microbiology (Reading). 1997 Mar;143 ( Pt 3):1029-1035. doi: 10.1099/00221287-143-3-1029. PMID: 9084185.

Lo M, Cordwell SJ, Bulach DM, Adler B. Comparative transcriptional and translational analysis of leptospiral outer membrane protein expression in response to temperature. PLoS Negl Trop Dis. 2009 Dec 8;3(12):e560. doi: 10.1371/journal.pntd.0000560. PMID: 19997626; PMCID: PMC2780356.

Line 239: The rationale for selection of the molecular weight range should be justified and a reference should be cited.

In this case we have cited previous publications, too:

Bugla-PÅ‚oskoÅ„ska, G.; Korzeniowska-Kowal, A.; Guz-Regner, K. Reptiles as a Source of Salmonella O48-Clinically Important Bacteria for Children: The Relationship Between Resistance to Normal Cord Serum and Outer Membrane Protein Patterns. Microb Ecol 2011, 61, 41–51, doi:10.1007/s00248-010-9677-7.

Futoma-Kołoch, B.; Godlewska, U.; Guz-Regner, K.; Dorotkiewicz-Jach, A.; Klausa, E.; Rybka, J.; Bugla-Płoskońska, G. Presumable Role of Outer Membrane Proteins of Salmonella Containing Sialylated Lipopolysaccharides Serovar Ngozi, Sv. Isaszeg and Subspecies Arizonae in Determining Susceptibility to Human Serum. Gut Pathog 2015, 7, doi:10.1186/s13099-015-0066-0.

Reeks BY, Champlin FR, Paulsen DB, Scruggs DW, Lawrence ML. Effects of sub-minimum inhibitory concentration antibiotic levels and temperature on growth kinetics and outer membrane protein expression in Mannheimia haemolytica and Haemophilus somnus. Can J Vet Res. 2005 Jan;69(1):1-10. PMID: 15745216; PMCID: PMC1142163.

It is unclear what the bottom half of figure 3 represents. For figure 2 - major tick marks for y axes are missing. 

Lines 276-277: We have finished the description to the Figure 3 with the explanation “The bottom half of the Figure 3 represents densitograms obtained with Image Lab. V. 4.1 (Bio-Rad, California, USA).”

In Figure 2 major marks have been added.

For figure 4 - A higher exposure blot will look more convincing. Please clarify the reason for the appearance of the box in the K3 area. Also clarify if there are positive and negative controls for binding in the dot blot. If not kindly comment or provide the necessary controls.

Figure 4 was corrected. According to your suggestion, we have carefully checked our repository of dot-blots and found more convincing figures showing clear signals for both C1q and C3c proteins. When it comes to exposure, this is the best quality that could be reached. In the legend to this figure, we have specified controls as follows: K1 - positive control 1; K2  positive control 2; K3 – negative control 1, instead of MPs PBS was used; K4, negative control 2 – instead of SPPP PBS was used.

That change cast the shadow on methods section 4.11, in lines 742-744 we have explained this matter: “Controls for binding in the dot blot were described below Figure 4. Since SS S. Typhimurium ATCC 14028 gave one of the highest signals in ELISA assay, it was regarded as a positive control.

For Table 3 and Figure 5 please explain why gene levels are looked at. What happens to the gene expression levels of the virulence associated genes? Please comment why a qPCR should or should not be performed. 

The aim of this part was to characterize strains in terms of possessing or not virulence genes. We proposed prolonged exposure to serum as a stress factor, which might initiate changes within the genome resulting in detection or not distinct genes using still the same primers as for parent strains. This is a first level of genetic analysis. Of course, it would be also interesting to use qPCR, but we feel, ultimately in a separate publication focused on changes in the expression of the genes connected to e.g. more sophisticated protein analysis and explaining biological mechanisms. In this paper, we have shown that using of conventional PCR protocols were sufficient to demonstrate possible change in fljB gene, what revealed in bacterial motility.

References to previous literature or results presented in the study should be provided for the statements made in lines - 530, 533, 816-818, 819-820.

Each statement you mention is supported by new citations as below:

Sharma D, Misba L, Khan AU. Antibiotics versus biofilm: an emerging battleground in microbial communities. Antimicrob Resist Infect Control. 2019 May 16;8:76. doi: 10.1186/s13756-019-0533-3. PMID: 31131107; PMCID: PMC6524306.

Carrascosa C, Raheem D, Ramos F, Saraiva A, Raposo A. Microbial Biofilms in the Food Industry-A Comprehensive Review. Int J Environ Res Public Health. 2021 Feb 19;18(4):2014. doi: 10.3390/ijerph18042014. PMID: 33669645; PMCID: PMC7922197.

B Drumm, A W Neumann, Z Policova, and P M Sherman. Bacterial cell surface hydrophobicity properties in the mediation of in vitro adhesion by the rabbit enteric pathogen Escherichia coli strain RDEC-1. J. Clin. Invest. Volume 84, November 1989, 1588-1594

Rosenberg M, Judes H, Weiss E. Cell surface hydrophobicity of dental plaque microorganisms in situ. Infect Immun. 1983 Nov;42(2):831-4. doi: 10.1128/iai.42.2.831-834.1983. PMID: 6642654; PMCID: PMC264505.

The idea of supporting of general statements encouraged us to improve the whole section Conclusions by adding appropriate numbers of tables or figures. Please have a look.

Discussion section title is missing. The manuscript needs extensive English editing to correct for grammatical errors.

The discussion section title had lost in the previous page. Now it is detached from the previous page.

Our manuscript had been corrected in English before submitting but you have right that there were some mistakes in the text. We hope we found them all and dismissed.

Reviewer 2 Report

Comments to the Authors

Manuscript ID: ijms-2080377

Title: The prolonged treatment of Salmonella enterica strains with human serum effects in phenotype related to virulence

The paper titled, " The prolonged treatment of Salmonella enterica strains with human serum effects in phenotype related to virulence ", is reasonably well-written and well-organized. 

The authors have presented the study in a systematic manner.

The authors thoroughly reviewed the literature for writing up this article.

The global message of the manuscript is clear and brings solid pieces of evidence and data.

The paper needs to be proof-read for the English language to avoid some mistakes for example: ”…life-threating ?- life threatening… ”; …in 28°C…?-…at 28°C…”; ”...this work an example…? ---…this work is an example….”.

This article can be an important consideration for many investigators and the information included in this paper needs to be published. I recommend the acceptance of this manuscript.

Author Response

Thank you for your compact and specific opinion. We are happy to see that you had a positive impression during reading our manuscript.

According to your suggestion:

The paper needs to be proof-read for the English language to avoid some mistakes for example: ”…life-threating ?- life threatening… ”; …in 28°C…?-…at 28°C…”; ”...this work an example…? ---…this work is an example….”.

we carefully checked the records and unified them properly, in details. Moreover, the final part of the introduction and some expressions in the whole text were linguistically corrected by dr Marcin Tereszewski. Please see a postscript in the Acknowledgements:

Additional thanks to Marcelina Wilczek for support in estimating CFU/mL-1 after each of eight passages of S. Isaszeg, S. Erlangen and S. Hammonia in 50% HS anddr. Marcin Tereszewski for English correction.